# The use of virtual nominal groups in healthcare research: An extended scoping review

Seung Ho Lee[1], Olle ten Cate[2], Michael Gottlieb[3], Tanya Horsley[4,5], Beverley Shea[5], Karine Fournier[6], Christopher Tran[7], Teresa Chan[8], Timothy J. Wood[9], Susan Humphrey-Murto[7,9]*

1 Faculty of Medicine, University of Ottawa, Ottawa, Canada, 2 Utrecht Center for Research and Development of Health Professions Education, Division of Education, University Medical Center Utrecht, Utrecht, The Netherlands, 3 Department of Emergency Medicine, Rush University Medical Center, Chicago Illinois, United States of America, 4 Royal College of Physicians and Surgeons of Canada, Research, Ottawa Ontario, Canada, 5 School of Epidemiology and Public Health, Faculty of Medicine, University of Ottawa, Ottawa, Ontario, Canada, 6 Health Sciences Library, University of Ottawa, Ottawa Ontario, Canada, 7 Department of Medicine, University of Ottawa, Ottawa Ontario, Canada, 8 Department of Medicine, Division of Emergency Medicine, Division of Education and Innovation, Faculty of Health Sciences, McMaster University, Hamilton, Ontario, Canada, 9 Department of Innovation in Medical Education, University of Ottawa, Ottawa, Ontario, Canada

* shumphrey@toh.ca

## Abstract

### Introduction

The Nominal Group Technique (NGT) is a consensus group method used to synthesize expert opinions. Given the global shift to virtual meetings, the extent to which researchers leveraged virtual platforms is unclear. This scoping review explores the use of the vNGT in healthcare research during the COVID-19 pandemic.

### Methods

Following the Arksey and O'Malley's framework, eight cross-disciplinary databases were searched (January 2020-July 2022). Research articles that reported all four vNGT stages (idea generation, round robin sharing, clarification, voting) were included. Media Synchronicity Theory informed analysis. Corresponding authors were surveyed for additional information.

### Results

Of 2,589 citations, 32 references were included. Articles covered healthcare (27/32) and healthcare education (4/32). Platforms used most were Zoom, MS Teams and GoTo but was not reported in 44% of studies. Only 22% commented on the benefits/challenges of moving the NGT virtually. Among authors who responded to our survey (16/32), 80% felt that the vNGT was comparable or superior.

**Data Availability Statement:** The entirety of the quantitative data collected and analyzed for this review is within the manuscript and its Supporting information files. Interview data is restricted due to

the participants only consenting to aggregate data sharing. Interview data will be made available upon request and approval from the University of Ottawa's Department of Innovation in Medical Education (DIME) at dime@uottawa.ca.

**Funding:** This project is being funded by a medical education research grant, Department of Medicine, University of Ottawa. The funders will not have any role in the study design, data collection, and analysis, decision to publish or preparation of the manuscript.

**Competing interests:** The authors have declared that no competing interests exist.

## Conclusions

The vNGT provides several advantages such as the inclusion of geographically dispersed participants, scheduling flexibility and cost savings. It is a promising alternative to the traditional in-person meeting, but researchers should carefully describe modifications, potential limitations, and impact on results.

## Introduction

Consensus group methods are often used to synthesize expert opinions when evidence is lacking or contradictory. They have been increasingly utilized in healthcare, business, engineering, and education [1–4].

These methods have been used to inform and evaluate a variety of healthcare-related activities (e.g. defining diagnostic criteria, informing management guidelines [2, 5, 6]; course evaluation [7]; amongst other uses [8–12]–Foundational principles of consensus methods include anonymity, iteration, controlled feedback, statistical group response, and structured interaction [1].

The NGT was developed for idea generation and group decision-making [3]. The structured format allows for effective generation and prioritization of ideas. The NGT has several key steps: presentation of the nominal question, silent generation of ideas in writing, round-robin sharing of ideas from participants, group discussion and clarification of each idea, followed by anonymous individual voting to rank priority areas. Feedback of results followed by more discussion and re-rating may occur [13, 14].

The NGT differs from other consensus techniques by employing an in-person meeting among 5–12 participants [15]. This is touted by many to be a strength, as it not only allows discordant ideas on topics of mutual interest to be freely expressed and synthesized, but it also affords the opportunity to explore any differences in opinions. Further, the collaborative feature of the NGT may increase ownership of research among stakeholders and enhance the potential of informing policy or practice [3]. However, the small number of participants is also regarded as a limitation, and the potential for dominant members to unduly influence group decision-making cannot be dismissed, even if NGT aims to counteract this effect in its structured procedure.

The coronavirus disease of 2019 (COVID-19) pandemic has fundamentally changed the way we work, learn, and conduct research [16] and NGT has not been exempt from the shift; many researchers transitioned the in-person NGT meeting to a virtual environment [17–20]. For instance, Nelson et al. (2022) employed 3 nominal groups to identify burnout strategies in resident physicians, one being the traditional in-person meeting followed by two of which were held virtually via Zoom [17]. Timmermans et al. (2022) conducted three virtual NGTs (vNGT) and provided recommendations on the transition to the synchronous, online environment, but their claims were drawn solely from the authors' subjective experiences [20].

There remains ambiguity regarding the extent to which other researchers have shifted the NGT to a virtual format, the types of platforms used, and modifications made to the technique. In addition to this exploratory inquisition, it is also important to consider whether researchers voiced any challenges, or perceived advantages or disadvantages. Taken collectively, these lessons learned would provide guidance for future users of vNGT.

The COVID-19 pandemic has supported an upsurge in information and communication technologies, which fundamentally changed how individuals interact. The Media

Synchronicity Theory (MST) is a conceptual framework that considers the effectiveness of information and communication technologies in facilitating group work [21]. Through the lens of MST, all communication activities are grouped into two simple processes: conveyance (transmission of new information to the receiver that enables the creation and revision of individual understanding of a problem) and convergence (mutual process that governs how individuals understand and negotiate a common ground for a problem), the latter of which may involve more authentic, rapid back and forth transmission of information. For the NGT, the idea generation phase would presumably require more conveyance, while the discussion and clarification phase emphasize more convergence. Both processes are necessary, but the proportion depends on the complexity and inherent characteristics of the research.

In summary, the NGT has been used in a variety of research settings, to inform important decisions. Although interaction between panel members is fundamental to the process, many in-person meetings have moved online due to the pandemic. At present, it is unclear to what extent the NGT has been undertaken virtually, and what implications this may have for the decision process.

## Objective

The overarching purpose of this study is to explore the use of the virtual Nominal Group Technique (vNGT) in healthcare research. Specific objectives are to answer the following questions: 1) To what extent has the NGT been used virtually? 2) What virtual communication platforms are used? 3) What modifications to the technique were made to accommodate this online format? And 4) What advantages and disadvantages were noted by authors?

## Methods

Following a study protocol that has been published [22], a scoping review was conducted. A scoping review was considered appropriate since this topic is poorly defined, and the purpose was to map the literature, find key concepts, types and sources of evidence and identify gaps in the literature. The study followed the Arksey and O'Malley framework [23] and the Preferred Reporting Items for Systematic Reviews and Meta-Analyses Extension for Scoping Review [24]. Written ethics approval was granted from the University of Ottawa Research Ethics Board on September 26, 2022. All participants answering the survey provided written informed consent.

### Step 1: Identifying the research question

Our study began with a broad objective: to explore the use of the vNGT in healthcare research. This study included all English-language published research in healthcare and healthcare education that used the NGT in a virtual format. This included using any non-in-person format, such as videoconferencing. The outcomes of interest included the author's description of how the vNGT was used, and the perceived success of the process, benefits, risks, and challenges. Informed by a preliminary search of the literature, we decided upon the following questions:

1. To what extent has the NGT been used virtually?

2. What virtual communication platforms were used?

3. What modifications to the technique were made to accommodate this online format?

4. What advantages and disadvantages were noted by authors?

**Table 1. Final inclusion and exclusion criteria.**

| Inclusion criteria | Exclusion criteria |
|---|---|
| English language | |
| Date limit: January 1, 2020 –July 15, 2022 | Conference proceedings, published abstracts, Reviews, Editorials, Opinion pieces |
| Full text articles | |
| Original research using the nominal group technique | |
| Nominal group technique must be described in sufficient detail (e.g. cannot simply mention "nominal group technique" with no further description) | |
| Must mention that all 4 key stages of the nominal group: idea generation, sharing of ideas, discussion/clarification and voting were completed and reported | |
| Could be one of several consensus methods used within a single study | |
| May include any research topic | |
| All stages of the nominal group technique were completed "virtually" (any of e-mail,online, any virtual platform, telephone) | |
| Can combine virtual and face-to-face for any given stage | |

## Step 2 & 3: Identifying relevant articles and article selection

We started with the following framework: Population, Concept, Context. The population included any published research studies using the nominal group technique, the concept entailed the use of virtual modalities to execute the nominal group technique, and context involved any study topic.

Several pilot searches were undertaken to define the search strategy. Details can be found in the published protocol [22]. Final inclusion and exclusion criteria are noted in Table 1.

The final search strategies were executed by an information specialist (KF) and peer reviewed using the PRESS guideline [25]. The searches were conducted July 15th, 2022, in: MEDLINE(R) ALL (OvidSP), Embase (OvidSP), CINAHL (EBSCOHost), ERIC (OvidSP), Education Source (EBSCOHost), APA PsycInfo (OvidSP), Web of Science, and Scopus to retrieve references published January 2020 to July 2022. This time period was selected to detect studies conducted during the COVID-19 pandemic to reflect the more rapid shift to virtual formats. No search filters, or language limits were used, but conference abstracts were removed since only full papers were of interest. The search strategies are included in S1 File.

Using the Covidence software [26], all titles and abstracts were reviewed in duplicate with varying pair combinations by three co- authors (SHL, SHM, MG and a research assistant). Those meeting inclusion criteria, or if not clearly conducted in person were pulled for full text review. Duplicate review of full text articles against inclusion/exclusion criteria was completed. All conflicts were resolved via consensus discussion with a third member of the team.

## Step 4: Charting the data

Through an iterative process the data extraction form was developed. In addition to publication-level information and demographics, concepts related to the MST were explored. Details can be found in the published protocol [22].

Following best practices from the manual for evidence synthesis [27] at least two members (SL, SHM) independently reviewed 20% of articles applying the final data extraction form. Thereafter, one member of the research team carried out data extraction with verification

from a second member. Any ambiguous items that arose were resolved through discussion with the senior author (SHM).

### Step 5: Collating, summarizing, and reporting the results

Both quantitative and thematic analyses were used to synthesize study results. Quantitative analysis focused on the nature (e.g., education, clinical research, guideline development) and distribution of relevant articles. Two members of the research team (SL, SHM) independently reviewed the data to identify preliminary themes as informed by the MST. Several group meetings with all team members were held to review the data and to agree on a final summary of findings.

### Step 6: Survey of the authors

Upon review of several studies, it became apparent that many of the articles did not comment on the "virtual" aspect of the NGT. As a result, we disseminated an online survey to the corresponding authors for articles included in the study. The survey sought to confirm which virtual platform was used and for which steps of the NGT, if additional functions were used (e.g., chats features) or modifications made to the NGT to accommodate the virtual platform, why the virtual platform was used, their general impressions and perceived benefits and challenges, comparing their experience with in-person NGTs, and any lessons learned. The survey is available in S2 File. The Survey. The Preferred Reporting Items for Systematic reviews and Meta-Analyses extension for Scoping Reviews (PRISMA-ScR) Checklist can be found in S3 File.

## Results

A total of 2,589 records were identified through database searching, of which 1,754 were removed as duplicates (Fig 1). Excluding irrelevant titles and abstracts, 598 full-text articles were assessed for eligibility based on our exclusion criteria. Resulting in 32 full-text articles meeting inclusion criteria.

### Study demographics

Demographic information can be found in Table 2. The largest number of studies were published in 2022 (50%; 16/32) [6, 28–42], followed by 2021 [43–54] and 2020 (13%; 4/32) [19, 55–57]. As seen in Table 2 geographic distribution, virtual studies were predominantly carried out at the national level (53%; 17/32), followed by intercontinental (19%; 6/32), international (16%; 5/32), and local settings (6%; 2/32).

Research questions were more related to healthcare (87.5%; 28/32) than to healthcare education (12.5%; 4/32). Within healthcare, topics were varied and included items such as improving care for inmate dementia [30], home rehabilitation for stroke survivors [49], COVID-19 vaccine rollout in pharmacies [51] barriers to testing lipids and achieving disease control in rheumatoid arthritis [56, 57]. Patients were often central to the vNGT with topics including patient views on the treatment of osteoporosis [28], multiple myeloma [50] or osteoarthritis [43] and patient reported outcomes in heart failure [33].

### vNGT participants

Five studies did not report the number of participants (15.6%) [6, 38, 41, 46, 47]. For those that did, the total number of participants per NGT group ranged from 2–20, however it was not always clear how many participants were in each NGT group as some studies ran several NGT groups in parallel (Table 2).

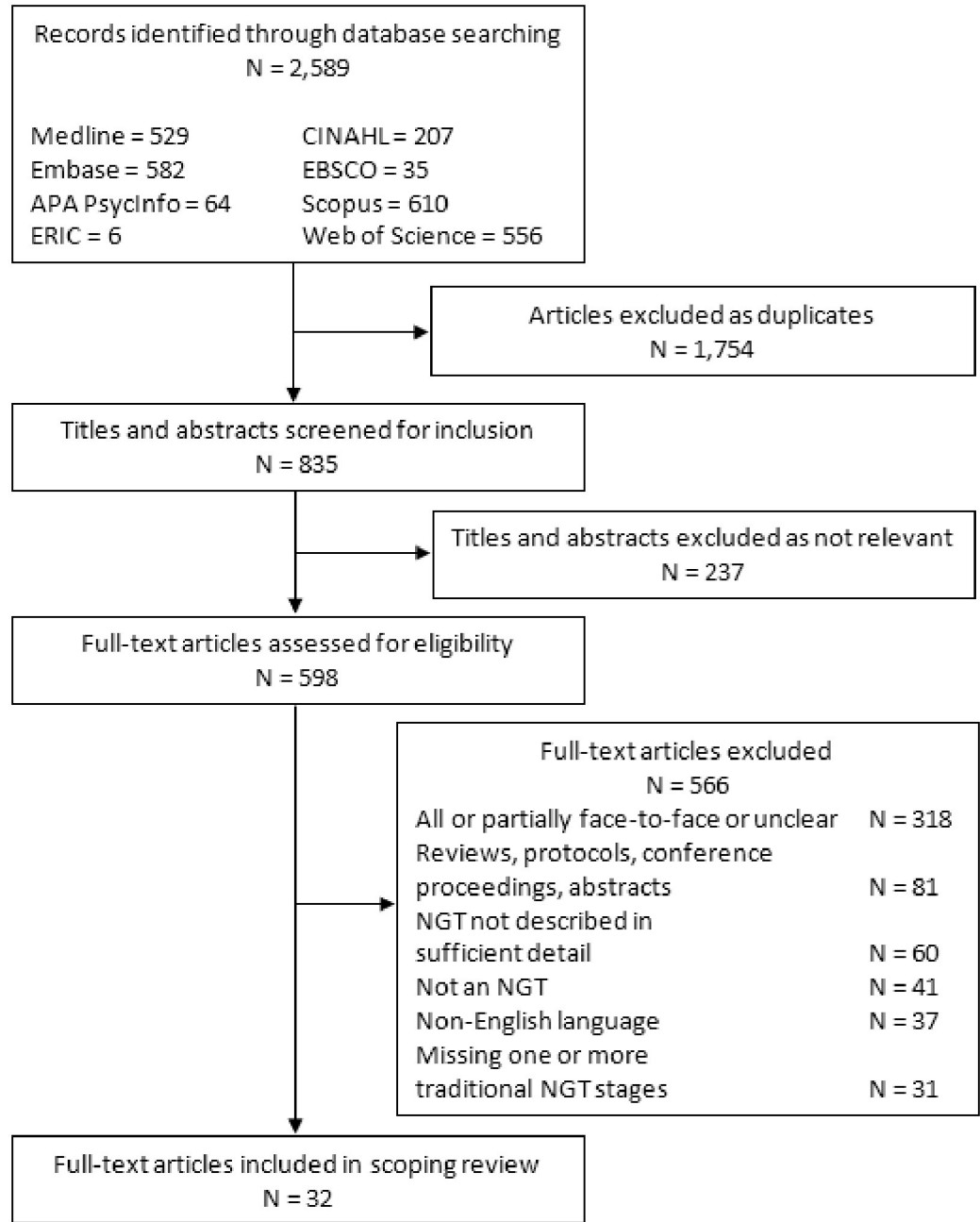

**Fig 1. PRISMA flowchart of the literature search and study selection process for a scoping review of virtual nominal group techniques published between March 2020 and July 2022.** Abbreviation: NGT, nominal group technique.

## Virtual modality

Table 3 provides details of the technique itself. All studies used an online conferencing modality, but many did not report which platform was used (43.7%; 14/32). Of those that did (n = 18), the most common software noted was Zoom (66.6%; 12/18), followed by Microsoft Teams (11.1%; 2/18), GoTo (11.1%; 2/18), Crisco (5.6%; 1/18) and Skype (5.6%%; 1/18). Half of the articles also added an email or e-survey modality to the study design (50%; 16/32) either

**Table 2. Demographic features of vNGT studies (n = 32).**

| First author | Study year | Journal | Topic | NGT used to generate… | Geographic distribution | Total participants | NGT groups | Participants per group |
|---|---|---|---|---|---|---|---|---|
| Al-Yateem [44] | 2021 | Journal of Nursing Management | Nursing recruitment | Interview questions for nursing recruitment | L* | 30 | 3 | NR |
| Aronson [45] | 2021 | Respiratory Research | Respirology | Physician perspectives on interstitial lung disease | N* | 25 | NR | NR |
| Bavelaar [29] | 2022 | Patient Education and Counseling | QI* –palliative care in dementia | Question prompts from family caregivers of dementia patients | IC* | 43 | NR | 4–20 |
| Beaudart [28] | 2022 | Archives of Osteoporosis | Osteoporosis | Patient-stated treatment preferences for osteoporosis | N | **27** | **3** | **9, 8, 10** |
| Choquette [46] | 2021 | Pharmacy Practice | Rheumatology | Monitoring drug algorithm for pharmacists on rheumatology | N | NR* | | |
| Dawson [47] | 2021 | European Urology Focus | Nocturia, endocrinology causes | Factors impacting nocturia care from primary care physicians | N | NR | NR | ≤6 |
| duToit [30] | 2022 | Gerontologist | QI—prison dementia | Recommendations for providing inmate dementia care | N | 27 | NR | NR |
| Evangelista [48] | 2021 | Military Medicine | Education—Ophthalmology curriculum development | Standardized curriculum for ophthalmological military training | N | 8 | NR | NR |
| Fisher [49] | 2021 | BMJ Open | QI—Stroke rehabilitation | Components of home rehabilitation for disabled stroke survivors | N | 12 | 2 | 6 |
| Hoops [31] | 2022 | Academic Medicine | Education | Priorities to inform creation of learning objectives for firearm injury education | N | 33 | NR | NR |
| Janssens [50] | 2021 | Frontiers in Medicine | QI—patient care | Patient-preferred characteristics of multiple myeloma treatments | I* | 24 | 4 | 6 |
| Johnson [19] | 2020 | International Journal of Environmental Research and Public Health | QI—school nutrition | Ideas for nutrition-guided school food provision model | N | Idea NGT: 21 consensus NGT: 11 | Idea NGT: 4 Consensus NGT: NR | Idea NGT: 8, 5, 5, 3 Consensus NGT: NR |
| Klaic [32] | 2022 | Implementation Science | QI—research waste | Implementable healthcare interventions to reduce research waste | IC | 7 | NR | NR |
| Lawson [33] | 2022 | European Journal of Cardiovascular Nursing | Cardiology (heart failure) | Patient-reported outcome measures to reduce heart-failure-related admissions | N | 11 | NR | NR |
| Lee [55] | 2020 | International Journal of Environmental Research and Public Health | Individuals with Disabilities | Barriers to healthy eating in mobility-impaired patients | N | 20 | 7 | NR |
| Li [34] | 2022 | International Journal of Environmental Research and Public Health | Mental Health and Lived Experiences | Perspectives of service workers on lived-experience research | N | 15 | NR | NR |
| Liem [35] | 2022 | Musculoskeletal Care | Education—Rheumatology and rehabilitation | Residency curriculum on primary care physical therapy for systemic sclerosis patients | N | 29 | 2 | NR |

*(Continued)*

**Table 2.** (Continued)

| First author | Study year | Journal | Topic | NGT used to generate... | Geographic distribution | Total participants | NGT groups | Participants per group |
|---|---|---|---|---|---|---|---|---|
| Love [36] | 2022 | Aphasiology | Speech and language | Perspectives of speech-language pathologists to develop a screening tool to identify cognitive communication disorder | IC | 5 | NR | NR |
| McClunie-Trust [37] | 2022 | International Journal of Qualitative Methods | Education/research | Experiences of research members involved in nursing education | I | 7 | NR | NR |
| Michel [51] | 2021 | International Journal of Clinical Pharmacy | QI—Vaccination in Pharmacies | Factors for COVID-19 vaccination programmes in community pharmacies | I | 23 | NR | 5–6 |
| Navarro-Millan [56] | 2020 | BMC Rheumatology | Rheumatology (hyperlipidemia in those with RA) | Rheumatologist and PCP perspectives on barriers to lipid testing among RA patients | N | Rheum: 27 PCP: 20 | Rheum: 3 PCP: 3 | Rheum: 11, 8, 7 PCP: 7, 4, 9 |
| Occomore-Kent [52] | 2021 | International Journal of Language and Communication Disorders | Speech and language | Speech-language pathologists' views of extending their role to work ENT patients | I | 9 | NR | NR |
| Ostbo [53] | 2021 | ARC Open Rheumatology | Rheumatology (scleroderma) | Resources for systemic sclerosis patients for nutrition and diet information | IC | 15 | 4 | NR |
| Owensby [57] | 2020 | Arthritis Care & Research | QI—RA control | Patient- and rheumatologist-perceived barriers to achieving disease control | N | Rheum: 25 Patients: 37 | NR | NR |
| Ridgway [6] | 2022 | European Urology Focus | Nephrology and endocrinology | Management of nocturia in chronic kidney disease in primary care physicians and non-nephrology specialists | N | NR | NR | 5–7 |
| Ryan [38] | 2022 | Implementation Science Communications | Implementation studies assessment tool | Tool development to describe implementation studies and assessing risk of bias of implementation outcomes | - | NR; "a small expert panel" | | |
| Singh [43] | 2021 | Arthritis Research & Therapy | Osteoarthritis | Patient views on the ineffectiveness of the current knee osteoarthritis treatments | L | 48 | NR | 2–8 |
| Singh [39] | 2022 | Rheumatology Advances in Practice | Rheumatology | Healthcare provider views on gout disease modification | IC | 20 | 6 | 5, 4, 3, 3, 3, 2 |
| Smith [40] | 2022 | BMC Health Services Research | QI—Deteriorating patients | Behaviour change techniques for nursing staff to manage deteriorating patients | N | 19 | 2 | 12, 7 |
| vanMerode [41] | 2022 | European Urology Focus | Urology | Management principles of nocturia and endocrine disease for primary care physicians | - | NR | NR | ≤6 |
| Volkmer [42] | 2022 | Disability and Rehabilitation | QI | Speech-language therapist interventions for people with primary progressive aphasia | I | 15 | NR | NR |
| Wood [54] | 2021 | Physiotherapy | Physiotherapy | Exercise treatment targets for patients with non-specific low back pain | IC | 39 (32 completed all stages) | 2 | 15 N, 24 I (12 N, 20 I completed all stages) |

*Acronyms: QI = Quality Improvement; L = local; N = national; I = International; IC = Intercontinental; NR = not reported

**Table 3. Select characteristics of 32 studies of virtual nominal groups conducted over January 2020 to July 2022.**

| First author | Study year | Parallel (P)* or mixed (M)* | Virtual modality used | | | Asynchronous stage(s) | | | Items provided | Discussion documentation | | Consensus defined a priori | Comments on using virtual? | Time (m) |
|---|---|---|---|---|---|---|---|---|---|---|---|---|---|---|
| | | | Online | Phone | Email/E-survey | IG# | RR# | DC# | | Recorded | Transcribed | | | |
| Al-Yateem [44] | 2021 | ✓(P)(M) | N/S†† | | ✓ | ✓ | | | ✓ | | | | ✓ | 90 |
| Aronson [45] | 2021 | | N/S | | | | | | | ✓ | | | | 40–140 |
| Bavelaar [29] | 2021 | ✓(P)(M) | N/S | ✓ | ✓ | ✓ | | ✓† | ✓ | | ✓ | | ✓ | - |
| Beaudart [28] | 2022 | | GoTo | | | | | | ✓ | ✓ | ✓ | | ✓ | 104 |
| Choquette [46] | 2021 | | Zoom | | | | | | | | | | | - |
| Dawson [47] | 2021 | | N/S | | | | | | | ✓ | ✓ | ✓ | | 120 |
| duToit [30] | 2022 | | Zoom | | ✓ | | | | | | ✓ | | | 60 |
| Evangelista [48] | 2021 | | N/S | | ✓ | ✓† | | | | | | ✓ | | 120 |
| Fisher [49] | 2021 | | Teams | | | | | | ✓ | | ✓ | ✓ | ✓ | 30 |
| Hoops [31] | 2022 | | Zoom | | ✓ | ✓ | | ✓† | ✓ | ✓ | ✓ | ✓ | | 240 |
| Janssens [50] | 2021 | ✓(P) | N/S | ✓ | | | | | ✓ | ✓ | ✓ | | ✓ | 90 |
| Johnson [19] | 2020 | | CiSCO | | ✓ | | | | | | ✓ | | | 120 |
| Klaic [32] | 2022 | | Zoom | | | | | | | | | | | - |
| Lawson [33] | 2022 | | N/S | | | | | | | ✓ | | | | - |
| Lee [55] | 2020 | | Zoom | | ✓ | | | | ✓ | | | | | 90 |
| Li [34] | 2022 | | Zoom | | ✓ | | | | | | ✓ | ✓ | | - |
| Liem [35] | 2022 | | Zoom | | ✓ | ✓ | | | ✓ | ✓ | ✓ | | | 90 |
| Love [36] | 2022 | | Zoom | | ✓ | ✓ | | | ✓ | ✓ | ✓ | | | 120 |
| McClunie-Trust [37] | 2022 | | Zoom | | | | | | | | ✓ | | | - |
| Michel [51] | 2021 | | Zoom | | ✓ | | | | | | ✓ | | ✓ | 120 |
| Navarro-Millan [56] | 2020 | | N/S | ✓ | | | | | | | ✓ | | | 90 |
| Occomore-Kent [52] | 2021 | ✓(P) | N/S | | | | | | | | ✓ | | | - |
| Ostbo [53] | 2021 | | GoTo | | ✓ | ✓ | | | | | ✓ | | | 90–120 |
| Owensby [57] | 2020 | | N/S | | ✓ | | | | | ✓ | ✓ | | ✓ | 60 |
| Ridgway [6] | 2022 | | N/S | | | | | | | ✓ | ✓ | | | 120 |
| Ryan [38] | 2022 | | N/S | | | | | | ✓ | | ✓ | | | - |
| Singh [43] | 2021 | | N/S | | | | | | | ✓ | ✓ | | | 90–120 |
| Singh [39] | 2022 | | Zoom | | | | | | | ✓ | ✓ | | ✓ | 60 |
| Smith [40] | 2022 | | Teams | | | | | | ✓ | | | | | 120 |
| vanMerode [41] | 2022 | | N/S | | ✓ | | | | | ✓ | ✓ | | | 120 |
| Volkmer [42] | 2022 | | Zoom | | ✓ | | | | | ✓ | ✓ | | | 90 |

*(Continued)*

**Table 3.** (Continued)

| First author | Study year | Parallel (P)* or mixed (M)* | Virtual modality used | | | Asynchronous stage(s) | | | Items provided | Discussion documentation | | Consensus defined a priori | Comments on using virtual? | Time (m) |
|---|---|---|---|---|---|---|---|---|---|---|---|---|---|---|
| | | | Online | Phone | Email/E-survey | IG# | RR# | DC# | | Recorded | Transcribed | | | |
| *Wood* [54] | 2021 | ✓(P) | Skype | | ✓ | ✓ | | | ✓ | | | | | 120 |

††N/S = not specified

*(P) denotes entire in-person session(s) held in parallel alongside virtual sessions;

†Indicates that a synchronous modality was also used in conjunction/addition to the asynchronous modality

(M) denotes in-person stage(s) mixed with rest of the virtual stages in a single session

#IG, RR, and DC refer to idea generation, round-robin sharing, and discussion/clarification, respectively

before, during, or after participants convened online. The rest used telephone communication in addition to online conferencing (9%; 3/32). Some studies held virtual sessions in parallel with in-person offerings to conduct the NGT simultaneously among various groups (16%; 5/32).

## vNGT modifications for virtual

Modifications to accommodate the technique virtually were often implied to have taken place, but rarely commented upon (25%; 8/32). Notably, the online break-out room functionality replaced physical meeting rooms to conduct multiple nominal group sessions [51]. Investigators also implemented online sheets [51], interactive whiteboards, and slideshows [39] to complement the process.

Asynchronous communication modalities, defined as non-interactive, non-simultaneous platforms such as email threads and e-survey, were noted in a quarter of the articles (25%; 8/32). The idea generation step was conducted asynchronously in 8 of 32 articles (25%). At the discussion and clarification stage, Bavelaar et al., 2022 [29] and Hoops et al., 2022 [31] offered asynchronous participation in conjunction with the interactive discussion in the form of email threads.

Five studies held in-person and virtual sessions in parallel, two of which allowed for both in-person and virtual within the same meeting based on context-dependent needs (e.g., geographic difference, member preference, technical challenges) [29, 31].

## vNGT method

A minority of studies defined consensus a priori (16%; 5/32). Four of the articles used a percentage score ranging from 50% to 75% agreement as the cut-off value [31, 48, 49, 54].

## Documentation

In many instances, the discussion phase of the NGT was formally documented either via audio or video recording (41%; 13/32) or transcription in the form of written notes during the meeting (69%; 22/32). Some opted to using both recording and live transcription (34%; 11/32).

## Other findings

Our inclusion criteria required that all stages of the vNGT be virtual (i.e., no in-person components). However, during the screening we did note that 52 articles incorporated at least one virtual component.

**Table 4. Benefits and challenges regarding NGT mentioned by authors in included studies (n = 32).**

| First author, year of study | Benefits and challenges regarding virtual adaptation | |
|---|---|---|
| | **Benefits** | **Challenges** |
| *Al-Yateem, 2021* [44] | • asynchronous online discussions allowed time to gather different perspectives and participants to fully understand the process by reviewing the material as many times as they wished<br>• online survey platform was a powerful tool to quickly prioritize the skills proposed by each participant. | • required extra effort from participants to support multiple meetings, asynchronous activities, and surveys<br>• results may have been compromised to an extent by directing participants to a priori information online |
| *Bavelaar, 2021* [29] | *None reported* | • differences in how and when group discussions were conducted may have influenced the results (pre-pandemic in some countries vs pandemic in others)<br>• adapting the group discussion from an onsite activity to a thread of emails, phone calls, or videoconferencing could have impacted the engagement process |
| *Beaudart, 2022* [28] | • afforded geographical flexibility<br>• webcam allowed for nonverbal communication<br>• no associated travel cost or facility rental fees<br>• safer than face-to-face meetings during waves of the pandemic | • limited generalizability to lower socioeconomic participants due to access to e-devices<br>• technical challenges prevailed despite premeeting checks<br>• inherent restricted sample size in online meetings precluded specific subgroup analysis (i.e., lack of power) |
| *Fisher, 2021* [49] | • allowed geographical distant members to be included<br>• the use of parallel groups also helped to provide insight into replicability of results | *None reported* |
| *Janssens, 2021* [50] | • telephone interviews allowed for more flexibility in choosing<br>• various dates for participation especially for older patients over online meetings | • participants who were not comfortable with online discussions or telephone (e.g., older participants) were less likely to participate, hence had to be given the choice to attend in-person |
| *Michel, 2021* [51] | • the completion and collation of the prepared excel sheets worked well<br>• the online nature of discussions enhanced sharing of ideas participants | • additional time for facilitators to compare and combine Excel sheets was not feasible due to differing time zones which led to perceived time constraints (only brief periods of time were possible for collation, clarification, and removal of duplicates) |
| *Owensby, 2020* [57] | • online participation offers scheduling flexibility, limits transportation barriers, and allows for a potentially more representative sample to provide feedback | • online participation may have acted as a barrier to participation |
| *Singh, 2022* [39] | *None reported* | • research studies using virtual NGT method are fewer, and more evidence is needed to be confident that this is analogous to in-person NGT |

*studies not listed did not comment on benefits or challenges of vNGT

## Descriptive characteristics

**Benefits and challenges of vNGT.** Only eight articles (25%;8/32) provided comments on the benefits or challenges (see Table 4). Online participation afforded geographical and scheduling flexibility, allowing a more representative sample of participants to convene in nominal group sessions [29]. One study suggested that the online nature of discussions may also have promoted the sharing of ideas among participants [51]. Janssens et al., (2021) suggested for older participants that the telephone was preferable [50]. The addition of asynchronous platforms was advantageous for one study as it provided more time for researchers to procure stakeholder perspectives, participants to understand study material through repeated review, and to efficiently prioritize and rank ideas.

In contrast, some comments directly addressed the challenges of adapting onsite activities to videoconferencing, email threads, and phone calls, which could have affected participant engagement. Two articles suggested that virtual modalities could have acted as a barrier to participation to some individuals, such as older participants [50, 57]. Prospective participants may

**Table 5. Survey of corresponding authors (n = 16): Summary of benefits, challenges, and key takeaways.**

**Benefits**

- Individualization of ideas due to lack of discussion among participants
- Technological restrictions (i.e., having to unmute) allow for structured sharing of ideas and discussion
- Better geographical representation, diversity, and accessibility of expert participants
- Reduced effort on logistical planning (i.e., money, time, and accommodation)
- Better engagement of quieter voices allowing for uniform contribution
- Ease of moderation and participation
- Platform capable of recording and transcription
- Higher attendance
- Enriched visualization of ideas

**Concerns**

- IChallenge for moderators and participants pre-pandemic as virtual platforms were not commonplace (i.e., mix-match of computers, phones, in-person)
- Technological limitations frequently present (i.e., Health IT access denial, positioning of devices to see physical flipchart in room)
- Akin to face-to-face in terms of not being able to reach consensus on the first NGT meeting
- Difficult to foster organic discussions
- Requiring of reminders to be silent during voting (no different than F2F)
- Greater confidentiality considerations
- Less opportunity to observe non-verbal cues
- More distractions from participant environment
- Not as engaging as F2F that may diminish opportunity for rich discussions

**Authors' recommendations**

- Clear communication of ground rules (i.e., ensuring confidentiality, secure log-on environment)
- Ample time to allow for preliminary introductions and participant access to platform
- Designation of IT support crew in addition to session moderator/facilitator and provision of contact information to participants
- Piloting and pre-meeting with participants prior to NGT
- Creative measures to facilitate each stage (i.e., private chat functions virtual whiteboards for sharing, online group maps for idea generation, email or surveys for voting, etc.)
- Consideration of issues with hybrid virtual-face-to-face sessions
- Shortening of session time
- Recruitment of an experienced moderator/facilitator
- Technical issues present in pre-pandemic era are mostly resolved as of current

have declined invitations to the study due to the extra effort required to partake in online meetings and asynchronous activities [44]. Furthermore, differences in geographical time zones posed time constraints for participants, which could have influenced group discussions, and therefore the results of the study [51]. One study mentioned that results could have been further compromised via asynchronous modalities as participants would be directed to a priori information online [44].

**Participant feedback.** One article openly gathered feedback from participants regarding the virtual nature of the NGT [44]. Participants expressed that the asynchronous format provided ample time to prepare ideas before of convening online.

**Corresponding author survey results.** The survey was completed by half of corresponding authors (50%; 16/32). Authors reported having been involved in in-person sessions 0 to 4 times in the past, and virtual sessions from 1 to 10 times. Participants were also asked their general impression of how the virtual worked compared to in-person NGT. Overall, 44% (7/16) felt that the vNGT was better, 36% (6/16) felt that both media performed similarly, and

19% (3/16) felt that vNGT was inferior to in-person NGT. Thematic analysis across the questions fell under benefits and concerns of using the vNGT as seen in Table 5.

## Discussion

This study set out to explore the use of the vNGT. Our findings demonstrate researchers use virtual platforms considering restrictions for travel and the requirements for social distancing. Although 8 databases were included in our study, the vast majority of articles reporting use of a virtual platform were from medical journals and related to healthcare.

The pandemic has imposed unprecedented sanctions in the way research is conducted, necessitating the transition from physical to virtual interactive modalities. Recent guidelines have suggested that researchers should report any modifications made as a result of extenuating circumstances (e.g., COVID-19), report important modifications to the methods, including mitigating strategies and impact on the results [58]. Our review demonstrated that most published research did not adhere to these guidelines; approximately a quarter of studies reported modifications to the method and just under a half which virtual platform was used. None considered the impact on outcomes. Overall, this is consistent with poor reporting seen in consensus research [4, 59, 60].

A few articles discussed the benefits and challenges of conducting the NGT virtually. Commonly cited benefits were increased accessibility for participants by overcoming barriers to participation, such as psychological discomfort in group settings [19], transportation issues [29, 49], or time constraints [32]. This advantage may have been amplified for healthcare professionals seeking to actively participate in research in the midst of battling the COVID-19 pandemic and has been previously reported [58]. However, lack of access or ability to use digital platforms may have reduced accessibility for some participants, due to socioeconomic disadvantage [28] or advanced age [50, 57]. Previous literature has certainly described age, race and literacy-related disparities in the use of technology with online populations being younger and more affluent [61, 62]. However, some report that online focus group participants were more likely to be non-white, less educated, and less healthy than the in-person sample [63]. Thus, online methods may lead to less representative participant demographics.

Several other limitations were noted such as timing of sessions if participants are from different time zones [51], and technical difficulties [28]. Testing the technology prior to use and having designated staff to address technical issues has been recommended [20], but is not always sufficient [28].

From the perspective of MST, the virtual platforms offered a variety of options to enhance communication. The capacity of communication modalities to support synchronicity is further informed by qualities intrinsic to the media, including, but not limited to transmission velocity, which refers to the speed at which a message reaches the recipient; parallelism, the number of simultaneous transmissions at any given time, and symbol sets, meaning different ways in which a message can be encoded. In general, symbol sets and transmission velocity enhance synchronicity, whereas parallelism decreases it [21, 64, 65].

The MST helped to inform our understanding of vNGT in pragmatic terms. Studies employed a diverse array of modalities depending on the NGT stage. In idea generation, which predominantly comprises one-way input, several studies [29, 31, 35, 36, 44, 48, 53, 54] adopted asynchronous modalities such as e-surveys or emails. On the other hand, discussion and clarification phases were conducted synchronously as demonstrated in all studies in our review with two also providing asynchronous additional options [29, 31]. This is in keeping with the idea that convergent communication benefits from higher transmission velocity. See Fig 2 for considerations at each stage.

| Stage of vNGT | Convergence (one way transmission of information - media that are intrinsically low in synchronicity) | Conveyance (involves a give-and-take transmission of messages thus requiring higher synchronicity) | Suggestions and future research considerations |
|---|---|---|---|
| Providing background information to participants and explaining process | Sufficient | Optional (provide brief summary during the meeting and answer any questions from participants) | Email information prior to the meeting may improve efficiency. |
| Idea Generation | Sufficient | Since generated in silence, should not be required. | Unclear if idea generation preceding the meeting is as effective as during the meeting. If completing without supervision will fewer items be generated? Should cameras remain on during this activity to encourage participants to stay focused on the task? |
| Round robin sharing | Possible for each participant to send ideas to the moderator before the meeting | Possible to share in round robin fashion during the meeting (as done within-person) | Unclear if participants generate more ideas while hearing others? Does the process increase social presence if conveyance considered. |
| Discussion and clarification | Possible, but not optimal | Should have high transmission velocity, high synchronicity, visual presence. | No studies have directly compared in-person compared to online high synchronicity including visual capacity. Does the virtual platform decrease social presence and engagement? |
| Anonymous voting | Possible to complete asynchronously | Can be done synchronously, on-line voting during meeting | Concern for participant attrition if done asynchronously. This has not been formally studied. |
| Feedback of results | Possible to complete asynchronously | Can be done synchronously, during meeting | Concern for participant attrition if done asynchronously. Benefit is less time spent in meeting. |
| Further discussion and/or voting | Possible to complete asynchronously | Can be done synchronously, during meeting | Concern for participant attrition if done asynchronously. Benefit is less time spent in meeting. |

**Fig 2. Media Synchronicity Theory—Considerations for vNGT and future research considerations.**

Concerns about engagement and interactivity were raised by several corresponding authors in our study, but none explored this in detail. Media richness theory suggests that computer-mediated group communication has lower social presence and less task focus [66] but this theory predates the current advances in platforms, which have improved our ability to see and interact with colleagues. One study that directly compared the web-based NGT to a traditional in-person format was using asynchronous technology with no visual capabilities [66]. Not surprisingly, the online participants were significantly less satisfied with the decision process compared to the traditional session, but only marginal differences were found in the outcomes [66]. Another study compared video to in person interviews. Interestingly Skype interviews yielded some of the most unguarded responses and richest data [67].

For this review, the vNGT was defined as having all stages in a virtual format. Several authors used a combination of virtual and in-person, but these were not included and are worthy of further exploration.

## Conclusion

The vNGT offers several potential benefits; inclusion of geographically dispersed participants, reduced time and expense for travel/meeting accommodations, and flexible scheduling. Reduced engagement and reduced participation for technology-challenge individuals are considerations. Since research on the vNGT itself is lacking, authors should clearly report modifications made and risks/benefits as well as potential impact on the consensus decision. Any minor adjustments in research protocols can introduce unanticipated vulnerabilities, thus compromising results and improved reporting is essential. In addition, more research is required to directly compare in-person to vNGTs including potential implications for cognitive biases [68].

## Supporting information

**S1 File. Search strategies.**
(DOCX)

**S2 File. The survey.**
(DOCX)

**S3 File. PRISMA SCR checklist.**
(DOCX)

## Acknowledgments

Dr. Humphrey-Murto would like to acknowledge the work of Amanda Pace and Kate Scowcroft from the Department of Innovation in Medical Education for assisting with the study.

## Author Contributions

**Conceptualization:** Seung Ho Lee, Olle ten Cate, Michael Gottlieb, Tanya Horsley, Beverley Shea, Karine Fournier, Christopher Tran, Teresa Chan, Timothy J. Wood, Susan Humphrey-Murto.

**Data curation:** Seung Ho Lee, Susan Humphrey-Murto.

**Formal analysis:** Seung Ho Lee, Olle ten Cate, Michael Gottlieb, Tanya Horsley, Beverley Shea, Christopher Tran, Teresa Chan, Timothy J. Wood, Susan Humphrey-Murto.

**Funding acquisition:** Seung Ho Lee, Olle ten Cate, Michael Gottlieb, Tanya Horsley, Beverley Shea, Karine Fournier, Christopher Tran, Teresa Chan, Timothy J. Wood, Susan Humphrey-Murto.

**Methodology:** Seung Ho Lee, Olle ten Cate, Michael Gottlieb, Tanya Horsley, Beverley Shea, Karine Fournier, Christopher Tran, Teresa Chan, Susan Humphrey-Murto.

**Project administration:** Susan Humphrey-Murto.

**Validation:** Seung Ho Lee, Susan Humphrey-Murto.

**Visualization:** Seung Ho Lee, Susan Humphrey-Murto.

**Writing – original draft:** Seung Ho Lee, Susan Humphrey-Murto.

**Writing – review & editing:** Seung Ho Lee, Olle ten Cate, Michael Gottlieb, Tanya Horsley, Beverley Shea, Karine Fournier, Christopher Tran, Teresa Chan, Timothy J. Wood, Susan Humphrey-Murto.

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
