## [Decision Letter · Decision Letter 0]

14 Nov 2023

PONE-D-23-20161The Use of Virtual Nominal Groups in Research: An Extended Scoping ReviewPLOS ONE

Dear Dr. Humphrey-Murto,

Thank you for submitting your manuscript to PLOS ONE. After careful consideration, we feel that it has merit but does not fully meet PLOS ONE’s publication criteria as it currently stands. Therefore, we invite you to submit a revised version of the manuscript that addresses the points raised during the review process.

We look forward to receiving your revised manuscript.

Kind regards,

Nabeel Al-Yateem, PhD

Academic Editor

PLOS ONE

Journal Requirements:

"Dr. Humphrey-Murto would like to acknowledge the work of Amanda Pace and Kate Scowcroft from the Department of Innovation in Medical Education for assisting with the study. This project was awarded a Department of Medicine Education Research Grant and a Faculty of Medicine Summer Studentship Award for Seung Ho Lee from the University of Ottawa."

"This project is being funded by a medical education research grant, Department of Medicine, University of Ottawa. The funders will not have any role in the study design, data collection, and analysis, decision to publish or preparation of the manuscript. "

Reviewers' comments:

Reviewer's Responses to Questions

**Comments to the Author**

1. Is the manuscript technically sound, and do the data support the conclusions?

Reviewer #1: Yes

2. Has the statistical analysis been performed appropriately and rigorously? 

Reviewer #1: N/A

3. Have the authors made all data underlying the findings in their manuscript fully available?

Reviewer #1: Yes

4. Is the manuscript presented in an intelligible fashion and written in standard English?

Reviewer #1: Yes

5. Review Comments to the Author

Reviewer #1: Thank you for the opportunity to review this scoping review. The authors mapped the literature with regards to the use of virtual Nominal Group Technique (NGT) in healthcare and health education research using a rigorous process. The manuscript is well-written and it was exciting reading through. I have a few comments for the authors.

Title: The authors may wish to modify the title slightly as follows: "The Use of Virtual Nominal Groups in Healthcare Research: An Extended Scoping Review". This is because in the methodology, it was stated that only studies that involved healthcare service, health education and guidelines were included. The addition of "healthcare" in the title increases the specificity of the topic.

Abstract: Fair

Introduction: The context was well described

Objectives: Just as stated in the title, you may wish to add "healthcare and health education" to the objectives as appropriate and the research questions at the methodology section.

Methodology: Clear. However, consider the following:

a. Expunge exclusion criteria 1. Since you included only studies that reported on the four key areas of NGT, this exclusion number 1 is not necessary.

b. How many authors participated in the covidence review?

Finally, this is a good paper and in my honest opinion should be considered for publication

6. PLOS authors have the option to publish the peer review history of their article (what does this mean?). If published, this will include your full peer review and any attached files.

Reviewer #1: No

---

## [Author Response · Author response to Decision Letter 0]

7 Feb 2024

Please review attached cover letter.

---

## [Decision Letter · Decision Letter 1]

4 Apr 2024

The Use of Virtual Nominal Groups in Healthcare Research: An Extended Scoping Review

PONE-D-23-20161R1

Dear Dr. Humphrey-Murto,

We’re pleased to inform you that your manuscript has been judged scientifically suitable for publication and will be formally accepted for publication once it meets all outstanding technical requirements.

Kind regards,

Nabeel Al-Yateem, PhD

Academic Editor

PLOS ONE

Additional Editor Comments (optional):

Reviewers' comments:

Reviewer's Responses to Questions

**Comments to the Author**

1. If the authors have adequately addressed your comments raised in a previous round of review and you feel that this manuscript is now acceptable for publication, you may indicate that here to bypass the “Comments to the Author” section, enter your conflict of interest statement in the “Confidential to Editor” section, and submit your "Accept" recommendation.

Reviewer #1: All comments have been addressed

2. Is the manuscript technically sound, and do the data support the conclusions?

Reviewer #1: Yes

3. Has the statistical analysis been performed appropriately and rigorously? 

Reviewer #1: N/A

4. Have the authors made all data underlying the findings in their manuscript fully available?

Reviewer #1: Yes

5. Is the manuscript presented in an intelligible fashion and written in standard English?

Reviewer #1: Yes

6. Review Comments to the Author

Reviewer #1: The authors have provided necessary responses to the comments raised by the reviewers. I consider their responses appropriate.

7. PLOS authors have the option to publish the peer review history of their article (what does this mean?). If published, this will include your full peer review and any attached files.

Reviewer #1: No

---

## [Editor Report · Acceptance letter]

17 May 2024

PONE-D-23-20161R1 

PLOS ONE

Dear Dr. Humphrey-Murto, 

I'm pleased to inform you that your manuscript has been deemed suitable for publication in PLOS ONE. Congratulations! Your manuscript is now being handed over to our production team.

Kind regards, 

on behalf of

Dr. Nabeel Al-Yateem 

Academic Editor

PLOS ONE